# Influence of Advanced Organ Support (ADVOS) on Cytokine Levels in Patients with Acute-on-Chronic Liver Failure (ACLF)

**DOI:** 10.3390/jcm11102782

**Published:** 2022-05-15

**Authors:** Leonard Kaps, Eva Maria Schleicher, Carolina Medina Montano, Matthias Bros, Simon Johannes Gairing, Constantin Johannes Ahlbrand, Maurice Michel, Pascal Klimpke, Wolfgang Maximilian Kremer, Stefan Holtz, Simone Cosima Boedecker-Lips, Peter Robert Galle, Daniel Kraus, Jörn M. Schattenberg, Christian Labenz, Julia Weinmann-Menke

**Affiliations:** 1Department of Internal Medicine I, University Medical Centre of the Johannes Gutenberg-University, 55131 Mainz, Germany; leonard.kaps@unimedizin-mainz.de (L.K.); eva.schleicher@unimedizin-mainz.de (E.M.S.); simonjohannes.gairing@unimedizin-mainz.de (S.J.G.); mail@constantin-ahlbrand.de (C.J.A.); maurice.michel@unimedizin-mainz.de (M.M.); pascal.klimpke@unimedizin-mainz.de (P.K.); maximilian.kremer@unimedizin-mainz.de (W.M.K.); stefan.holtz@unimedizin-mainz.de (S.H.); simonecosima.boedecker-lips@unimedizin-mainz.de (S.C.B.-L.); peter.galle@unimedizin-mainz.de (P.R.G.); daniel.kraus@unimedizin-mainz.de (D.K.); joern.schattenberg@unimedizin-mainz.de (J.M.S.); christian.labenz@unimedizin-mainz.de (C.L.); 2Cirrhosis Centre Mainz (CCM), University Medical Centre of the Johannes Gutenberg-University, 55131 Mainz, Germany; 3Department of Dermatology, University Medical Centre Mainz, Langenbeckstrasse 1, 55131 Mainz, Germany; gmedinam@students.uni-mainz.de (C.M.M.); mbros@uni-mainz.de (M.B.); 4Metabolic Liver Research Program, I. Department of Medicine, University Medical Centre of the Johannes Gutenberg-University, 55131 Mainz, Germany; 5Research Center for Immunotherapy (FZI), University Medical Centre of the Johannes Gutenberg-University, 55131 Mainz, Germany

**Keywords:** liver dialysis, cirrhosis, liver failure, hyperinflammation, extracorporeal liver support systems, albumin dialysis

## Abstract

Background: ADVanced Organ Support (ADVOS) is a novel type of extracorporeal albumin dialysis that supports multiorgan function in patients with acute-on-chronic liver failure (ACLF). No data exist on whether ADVOS affects inflammatory cytokine levels, which play a relevant role in ACLF. Aim: Our aim was to quantify cytokine levels both before and after a single ADVOS treatment in patients with ACLF at a regular dialysis ward. Methods and results: In this prospective study, 15 patients (60% men) with ACLF and an indication for renal replacement therapy were included. Patient liver function was severely compromised, reflected by a median CLIF-consortium ACLF score of 38 (IQR 35; 40). Blood samples were directly taken before and after ADVOS dialysis. The concentration of cytokines for IL-1β, IFN-α2, IFN-γ, TNF-α, MCP-1, IL-6, IL-8, IL-10, IL-12p70, IL-17A, IL-18, IL-23, IL-33 were quantified via a cytometric bead array. We found no significant (*p* > 0.05) change in cytokine levels, even when patients were stratified for dialysis time (<480 min versus ≥480 min). The relevance of the assessed cytokines in contributing to systemic inflammation in ACLF was demonstrated by Ingenuity pathway analysis^®^. Conclusion: Concentrations of pathomechanistically relevant cytokines remained unchanged both before and after ADVOS treatment in patients with ACLF.

## 1. Introduction

Liver cirrhosis causes more than 1.3 million deaths per year and imposes a substantial health burden both in the western world and around the globe [1]. Patients with liver cirrhosis are complexly ill, and are at high risk of developing life-threatening complications [2]. In addition to bleeding events, acute-on-chronic liver failure (ACLF) is one of the most life-threatening complications in cirrhosis and occurs at a frequency of 20.1 per 1000 person-years [3]. The European Association for the Study of Chronic Liver Failure (EASL-CLIF) consortium defined ACLF as acute decompensation of the cirrhotic liver with (multi-)organ failure [4]. As both single and multiorgan failure can occur, symptoms range from acute kidney injury (AKI), compromised hepatic synthetic function with hyperbilirubinemia, coagulopathy, and hepatic encephalopathy (HE), to severe systemic inflammation [4].

Inflammation is a physiological response of the immune system to pathogens, danger signals, and cell damage. Under normal conditions, inflammation occurs in a rapid but tightly controlled manner. Pro- and anti-inflammatory cytokines orchestrate the inflammatory process to facilitate an effective immune response that quickly resolves once a given pathogen is cleared [5].

Patients with liver cirrhosis suffer from chronic inflammation with elevated inflammatory markers, such as C-reactive protein (CRP), and leukocytosis [6]. In patients with decompensated cirrhosis, the balance between pro- and anti-inflammatory cytokines shifts toward an inflammatory state, resulting in strong immune activation [5]. In ACLF, this imbalance is exacerbated, leading to hyperactive systemic inflammation, causing further organ damage (Figure 1).

Except for liver transplantation (LTX), there is no specific treatment for ACLF, and clinical management relies on controlling complications. Extracorporeal albumin dialysis (ECAD) is an appealing concept for the treatment of ACLF [7]. It eliminates both protein-bound and water-soluble toxins and metabolites, mimicking liver and kidney detoxification [7]. Previous ECAD systems required large amounts of human albumin (HA). ADVanced Organ Support (ADVOS) consumes only minimal amounts of HA by efficiently recycling this natural, but scarce, product [8]. We previously demonstrated that ADVOS is both a safe and efficient procedure for removing liver-related toxins and metabolites in patients with ACLF [9].

The effect of ADVOS on cytokine levels has not yet been evaluated. We therefore sought to clarify whether ADVOS eliminates inflammatory cytokines in patients with ACLF.

## 2. Materials and Methods

### 2.1. ADVanced Organ Support (ADVOS)

We previously described in detail that ADVOS is an approved ECAD system that facilitates combined liver and kidney support (Figure 2) [9]. In brief, the setup consists of three communicating circuits. The first circuit circles blood through a conventional double-lumen dialysis catheter and a high flux dialysis membrane to eliminate water-bound toxins. In the second circuit, a dialysate containing a 2–4% albumin solution runs parallel to the first circuit, separated only by a semipermeable membrane when running through the dialyzer, loading the protein solution with protein-bound toxins and eliminating water-soluble toxins. In the third circuit, toxin-loaded albumin is detoxified. Here, the dialysate separates into two branches, where either acid or base is added. Due to the pH change and a lower temperature of around 28 °C, the albumin changes its conformation and releases the bound toxins, which are subsequently removed by a tertiary filtration process. Physiological conformation of albumin is re-established when the acidified and alkalized albumin dialysates pool, and the dialysates warm up to body temperature. The purified albumin is then ready for the next cycle (Figure 2).

According to our clinical standards, heparin was omitted. Anticoagulation was performed using citrate, independent of the patient’s prothrombin time to reduce the risk of bleeding. Patients received ADVOS as a discontinuous treatment in a regular ward. No additional renal replacement therapy was performed.

### 2.2. Study Design and Endpoint

In this prospective study, the blood samples of 18 patients with ACLF receiving ADVOS were collected from April 2019 to September 2021 at the I. Department of internal medicine, University Medical Center Mainz, Germany. The aim of this study was to assess the quantitative change in cytokine levels in blood sera from patients both before and after ADVOS treatment. Medical history and laboratory data were retrieved by chart review, or from the local laboratory system.

### 2.3. Inclusion and Exclusion Criteria

Patients were selected based on the following criteria (Table 1):

### 2.4. ADVOS Treatment Parameters

Treatment parameters of ADVOS were retrieved using a chart review. Data were incomplete for two patients. In total, patients received a median of 5 (IQR 3.3; 8) ADVOS cycles as a discontinuous treatment in a peripheral ward. Vasopressor therapy was discontinued after patients were subjected to the first ADVOS cycle. Patients received red blood cell transfusion or intravenous volume substitution with balanced electrolyte crystalloid solution during the treatment when indicated. On the day of sampling, the median duration of treatment was 480 min (IQR 360; 480). The median ultrafiltration rate was 120 mL/h (IQR 38; 225), with a median total ultrafiltration volume of 743 mL (IQR 358; 1204), and a median blood flow rate of 150 mL/h (IQR 115; 150). The dialysis flow was set to 320 mL/min.

### 2.5. Quantification of Cytokine Levels by Cytometric Bead Array

After sampling, blood was centrifuged to obtain sera as the supernatant. Afterward, sera were diluted 1:2 with a balanced electrolyte crystalloid solution and stored at −20 °C. On the day of analysis, diluted sera were thawed, and cytokine concentrations for IL-1β, IFN-α2, IFN-γ, TNF-α, MCP-1, IL-6, IL-8, IL-10, IL-12p70, IL-17A, IL-18, IL-23, and IL-33 were determined using a LEGENDplex™ Human Inflammation Panel 1 (13-plex) as recommended by the manufacturer’s protocol instructions (Biolegend, San Diego, CA, USA) [10,11,12,13]. In brief, samples were mixed with cytokine-specific capture beads and subsequently incubated first with detection antibodies and then with PE-conjugated detection antibodies (all at room temperature in the dark under shaking), and subjected to flow cytometric analysis. Results were analyzed using Qognit Legendplex Analysis Software (Version 7.1, Biolegend, San Diego, CA, USA).

### 2.6. Ingenuity Pathway Analysis (IPA^®^)

IPA^®^ predicts (de)activation of a disease’s related pathways based on a comprehensive database derived from experimental data [14]. Selected cytokines were linked to the inflammasome pathway, which is known to be a major driver of hyperinflammation in ACLF [6]. IPA^®^ predicted the regulatory effect of the linked cytokines on other relevant cytokines and the inflammasome pathway based on curated experimental data.

### 2.7. Statistical Analysis

Quantitative data are expressed as medians with interquartile ranges (IQRs). Categorical variables are given as frequencies and percentages. The Mann–Whitney U-test was used to test for any differences in variables. Our complete data analysis was exploratory. Hence, no adjustments for multiple testing were performed. For all tests, we used a 0.05 level to define statistically relevant deviations from the respective null hypothesis; however, due to the large number of tests, the *p*-values should be interpreted with caution and only as descriptive. Data were analyzed using GraphPad Prism Version 8.0.2 (GraphPad Software, San Diego, CA, USA).

## 3. Results

### 3.1. Baseline Characteristics 

In the final analysis, 15 patients with ACLF were included in the study, while 3 patients with either no evidence of cirrhosis or having incomplete chart review data were excluded (Appendix A). The majority of patients (60%) were men, with a median age of 50.5 years (IQR 42.2; 56.5). The main etiology for cirrhosis was excessive alcohol consumption (73%), followed by viral hepatitis (13%). All included patients had a diagnosis of ACLF with a median CLIF-C ACLF score of 52 (IQR 48; 56) and a high median MELD score of 38 (IQR 34.8; 40). Infections were identified in 40% of patients as the trigger for ACLF, whereas the cause remained obscure for five patients (33%). All patients had an indication for renal replacement therapy due to either critical laboratory parameters or fluid overload. Liver function was significantly compromised, as reflected by a high median bilirubin level of 23.4 (IQR 17; 34.7) and a spontaneous median INR of 1.9 (IQR 1.6; 4.6). Patients exhibited signs of hyperinflammation syndrome, as indicated by an elevated median WBC count of 15/nL (IQR 7.3; 18.9) and median CRP of 43 (IQR 25; 65) (Table 2).

Patients were severely ill, reflected by a high one-month mortality of 46% (seven patients died). Two patients received salvage LTX within one month. During hospitalization, patients received a median of five (IQR 3.3; 8) ADVOS cycles, and none were treated with vasopressors once ADVOS treatment began (Table 3).

### 3.2. Concentration of Cytokines before versus after ADVOS

Blood samples of patients were directly taken both before and after ADVOS treatment. Concentrations of inflammatory cytokines were quantified by a cytometric bead array. Cytokine levels were not significantly affected by ADVOS, even when patients were stratified for median treatment time (<480 min versus ≥480 min dialysis time; Figure 3 and Appendix A).

In line with cytokine levels, the standard inflammatory marker CRP remained unchanged by the treatment (Appendix A).

Furthermore, the detoxication efficiency of the ADVOS treatment for bilirubin and regular urinary substances is shown in Appendix A.

### 3.3. Relevance of Cytokines in ACLF 

Because cytokines are the major mediators of inflammation, we sought to clarify the relevance of previously assessed cytokines in ACLF through Ingenuity pathway analysis (IPA^®^) [5,14]. The key ACLF cytokines IL-10, IL-6, and TNF-α were linked to the inflammasome pathway, which is known to be a relevant pathway for sustaining systemic inflammation in ACLF. Based on curated experimental data, IPA^®^ predicted that IL-10 has an antagonistic effect on the inflammatory cytokines IL-6 and TNF-α and acts as an inhibitor in the cascade of the inflammasome pathway, inhibiting toll-like receptor 4 (TLR4) and caspase 8 (casp8) (Figure 4). For example, IL-10 decreases the expression of cytokines TNF-α and IL-6, while also directly inhibiting expressions of toll-like receptor 4 (TLR-4) and inflammatory casp8 [15,16,17,18,19,20,21,22]. IL-10 thus has a strong immunomodulatory effect both on key inflammatory cytokines and the inflammasome pathway.

## 4. Discussion

ACLF is a devastating complication in liver cirrhosis and predominantly occurs in younger patients with end-stage liver disease. It is associated with high short-term mortality, and clinical management of ACLF relies on symptomatic treatment when patients do not qualify for LTX. ADVOS is a novel ECAD system that supports multiorgan function and holds promise for bridging patients to LTX or possibly to their recovery. ADVOS proved to be safe for removing protein-bound and water-soluble toxins and metabolites in patients with ACLF, consuming only minimal amounts of HA [9]. However, no data exist on whether ADVOS improves mortality of patients with ACLF, necessitating randomized clinical trials.

Presently, no information is available on whether ADVOS affects cytokine levels in patients with ACLF. Cytokines are small proteins, ranging from 5 to 25 kDA, measuring approximately three-fold smaller than albumin, the smallest blood serum protein (Appendix A). Cytokines orchestrate inflammation and play an important role in systemic inflammation in ACLF [5,6]. Although albumin is not filtered, cytokines with heterogenous electrostatic properties and size could be eliminated by ADVOS (Appendix A).

We quantified cytokine levels both before and after ADVOS treatment in the blood sera of patients with ACLF. Cytokine levels were not found to be significantly affected by ADVOS, regardless of the type of cytokine and duration of dialysis (Figure 3 and Appendix A). This is an interesting finding because regular hemodialysis is known to induce the elevation of inflammatory markers [23]. The cause of inflammation in hemodialysis is multifactorial, and may originate from several sides. Repetitive contact of blood mononuclear cells with dialysis tubes and dialyzer membranes having insufficient biocompatibility may induce inflammation [23,24,25]. Complement activating membranes (e.g., cuprophan made membranes) induced the expression of IL-1β and of complement factor 5a (C5a) [23,26]. Dialyzer membranes affect cytokine levels either by activation of leukocytes or absorption of cytokines [27,28]. This is especially the case with modern manufactured polymer-based membranes that adsorb pro- and anti-inflammatory cytokines, resulting in a net balance of zero for the immune system [29]; However, it appears plausible that dialyzer membranes remove cytokines only to a minor extent due to their restrictively small surface area [30]. In addition to immune activation via interference with foreign materials, dialysate impurities (e.g., lipopolysaccharides of Gram bacteria) have been suspected of influencing body cytokines [31]; However, clinical data were heterogenous, mostly because dialysates referenced as “impure” were already of sufficient quality [32].

One of the hallmarks of ACLF is hyperinflammation, which is mainly driven by inflammatory cytokines [33]. We performed an IPA^®^-powered network analysis based on curated experimental data to determine the effects of the quantified cytokines on the ACLF inflammasome pathway [14]. We selected this pathway because of its relevance for hyperinflammation in ACLF [6]. IL-10 was identified as a main immunoregulatory cytokine in the pathway. IL-10 inhibits the key inflammatory cytokines IL-6 and TNF- α, and targets receptors and molecules in the pathway’s cascade (Figure 4). Because ADVOS does not change the concentration of pathomechanistically relevant cytokines (as demonstrated for IL-10) in ACLF, there is no indication that hyperinflammation might be aggravated.

In this study, several limitations need to be acknowledged. The interpretation of data was limited due to the small number of included patients and the lack of a matched control group with hemodialysis. Cytokine levels were determined after only one treatment, and possible cumulative dialysis effects occurring from performing ADVOS on consecutive days could not be evaluated. Due to the small sample size, the predictive value of quantified cytokines for, e.g., mortality, could not be defined.

In conclusion, we report, for the first time, that ADVOS does not change the concentration of pathomechanistically relevant cytokines in patients with ACLF after a single treatment.

## Figures and Tables

**Figure 1 jcm-11-02782-f001:**
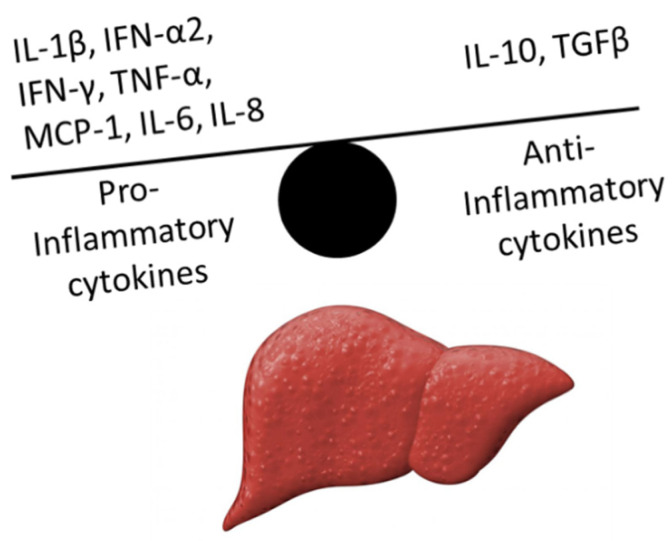
Cytokine disbalance in acute-on-chronic liver failure (ACLF). In ACLF, proinflammatory cytokines outnumber anti-inflammatory cytokines, giving rise to systemic inflammation (IL-1β, interleukin 1 beta; IFN-α2, interferon alpha-2; IFN-γ, interferon γ; TNF-α, tumor necrosis factor-α; MCP-1, monocyte chemoattractant protein 1; IL-6, interleukin 6; IL-8, interleukin 8; IL-10, interleukin 10; TGFβ, transforming growth factor β).

**Figure 2 jcm-11-02782-f002:**
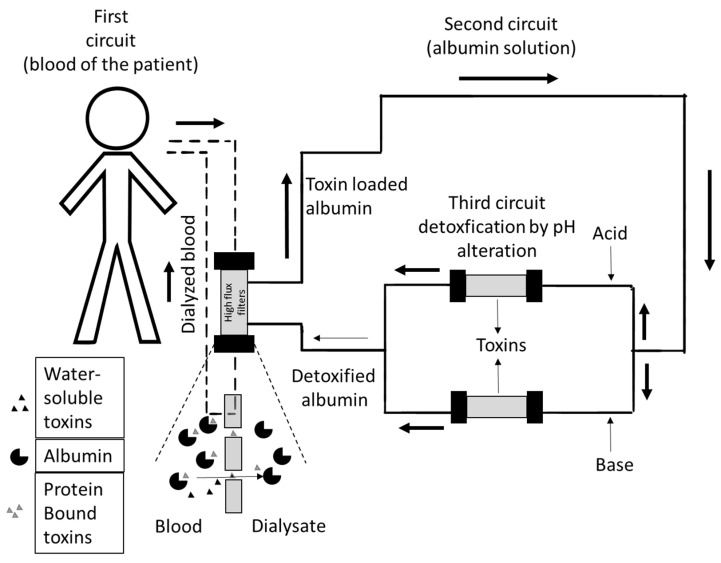
Schematic presentation of ADVanced Organ Support (ADVOS). In the first circuit, blood passes through a high flux filter that filters protein-bound and water-soluble toxins. In the second circuit, the toxin loaded albumin, together with water-soluble toxins, travel to the third circuit, where albumin is recycled by pH alteration. Base and acid are added, inducing a conformational change of albumin that consequently releases the toxins. Water-soluble toxins and released toxins are filtered out. When the acidified and basified albumin solution reunite, physiological pH is re-established and albumin regains its previous conformation, thus being recycled for the next cycle.

**Figure 3 jcm-11-02782-f003:**
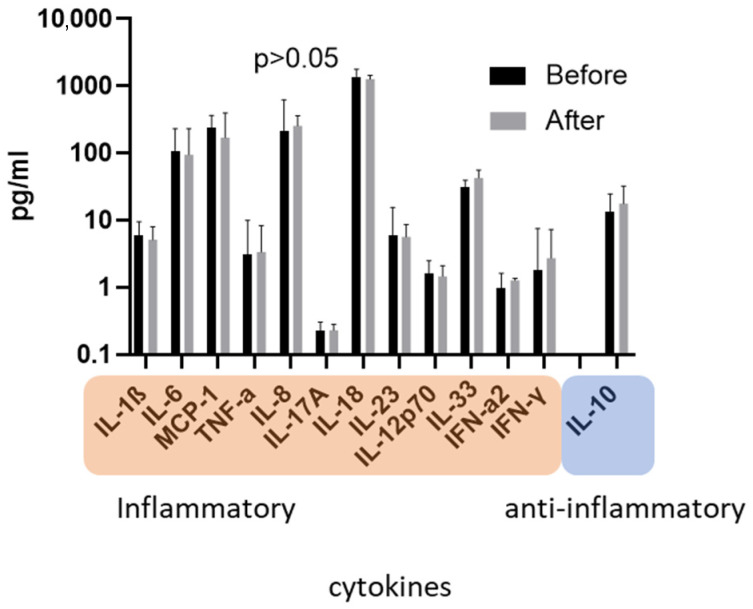
Concentrations of inflammatory and anti-inflammatory cytokines of blood samples taken from ACLF patients before and after ADVOS treatment as assessed by a cytometric bead array (median concentration, error bars indicate IQR).

**Figure 4 jcm-11-02782-f004:**
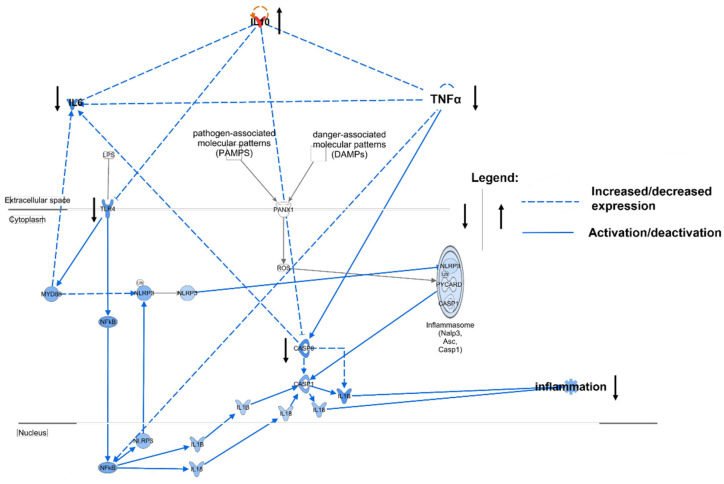
Inflammasome pathway relevant to ACLF predicted by IPA^®^. IL-10 was identified as the main immunomodulatory cytokine in the pathway, inhibiting the inflammatory cytokines IL-6 and TNF-α (dotted lines indicate increased or decreased expression, while solid lines show activation; Asc, apoptosis-associated speck-like protein containing a CARD; CASP, caspase; IL, interleukin; LPS, lipopolysaccharides; MCP-1, monocyte chemoattractant protein 1; MyD88, myeloid differentiation primary response 88; NFκB, nuclear factor kappa-light-chain-enhancer of activated B cells; NLRP3; NLR family pyrin domain containing 3; PANX1, pannexin 1; ROS, reactive oxygen species; TLR-4, roll-like receptor 4; TNF-α, rumor necrosis factor-α).

**Table 1 jcm-11-02782-t001:** Inclusion and exclusion criteria.

**Inclusion**	
1.	Clinical or histological evidence of liver cirrhosis *
2.	Acute decompensation indicated by ascites (II–III), deterioration of laboratory parameters or hepatic encephalopathy (West Haven criteria grade ≥ I) in line with CLIF consortium organ Failure Score ≥ 1
3.	Bilirubin ≥ 3 mg/dL and sudden prothrombin time-INR > 1.4
4.	HRS-AKI with anuria/oliguria diagnosed according to the EASL HRS/ACLF guidelines [4]
**Exclusion**	
1.	Mean arterial pressure ≤ 50 mmHg despite volume expansion
2.	Patients aged <18 years

* Diagnosis of liver cirrhosis was made using histology, typical appearance in ultrasound or radiological imaging, endoscopic features of portal hypertension, and medical history; ACLF, acute-on-chronic liver failure; CLIF, chronic liver failure; INR, International Normalized Ratio. HRS-AKI, hepatorenal syndrome-acute kidney injury; EASL, European Association for the Study of the Liver.

**Table 2 jcm-11-02782-t002:** Baseline characteristics of included patients.

**Total Patients, *n* (%)**		15 (100%)
	Male, *n* (%)	9 (60%)
	Age (years), median (IQR)	50.5 (42.2; 56.5)
	Type of liver failure (%)	ACLF (100%)
**Etiology of Cirrhosis**		
	Alcohol, *n* (%)	11 (73%)
	Viral hepatitis, *n* (%)	2 (13%)
	Cholestatic/Autoimmune, *n* (%)	1 (7%)
	Other/mixed, *n* (%)	1 (7%)
**ACLF Trigger**		
	Unknown, *n* (%)	5 (33%)
	Infections, *n* (%)	6 (40%)
	Varical bleeding, *n* (%)	4 (27%)
**Liver Function**		
	Child-Pugh score, *n* (%)	3 B (20%), 12 C (80%)
	MELD, median (IQR)	38 (35; 40)
**CLIF-C ACLF Calculator, Median**		
	CLIF-C ACLF score (IQR)	52 (48; 56)
	ACLF grade (IQR)	15 (2; 3)
	CLIF Organ Failure Score (IQR)	11 (11; 12)
	Liver failure (IQR)	3 (3; 3)
	Kidney failure (IQR)	3 (3; 3)
	Cerebral failure (IQR)	2 (1; 2)
	Coagulation failure (IQR)	1 (1; 3)
	Circulatory failure, *n* (IQR)	1 (1; 1)
	Lung failure, *n* (IQR)	1 (1; 1)
	One-month probability of dying (IQR)	33 (24; 44)
**Laboratory, Median**		
	Sodium, mmol/L (IQR)	136 (130; 143)
	Potassium, mmol/L (IQR)	4 (3; 5.2)
	BUN, mg/dL (IQR)	43 (35; 71.3)
	SCr, mmol/L (IQR)	2.8 (2.3; 4.6)
	INR, median (IQR)	1.9 (1.6; 4.6)
	Bilirubin, mg/dL (IQR)	23.4 (17; 34.7)
	CRP, mg/L (IQR)	43 (25; 65)
	Albumin, g/L (IQR)	20 (17; 31.3)
	Thrombocytes, count/nL (IQR)	87 (54; 111)
	WBC, count/nL (IQR)	15 (7.3; 18.9)

ACLF, acute-on-chronic liver failure; BUN, blood urea nitrogen; CLIF, consortium organ failure score; MELD, model of end-stage liver disease; INR, International Normalized Ratio; IQR, interquartile range; WBC, white blood cell count; CRP, C-reactive protein; SCr, serum creatinine.

**Table 3 jcm-11-02782-t003:** Treatment details and outcome of included patients.

**Treatment, Median**		
	Dialysis time at blood sampling, minutes (IQR)	480 (360; 480)
	Total ADVOS cycles during hospitalization, (IQR)	5 (3.3; 8)
	Vasopressor therapy	0
**Outcome, *n* (%)**		
	Liver transplantation	2 (13%)
	One-month mortality	7 (46%)
	Overall mortality	8 (53%)

## Data Availability

The datasets generated during and/or analyzed during the current study are available from the corresponding author upon reasonable request.

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
