# Peer review of "Influence of Advanced Organ Support (ADVOS) on Cytokine Levels in Patients with Acute-on-Chronic Liver Failure (ACLF)"

_jcm, 2022, doi:10.3390/jcm11102782_

Round 1

Reviewer 1 Report

The project is interesting. However, they must show how innovative the equipment is and the advantage of albumin dialysis, which allows a high performance of detoxification. After detoxification of the liver and the passage of the blood through the filters, the levels of cytokines are not reduced.
I have some suggestions and clarifications for the authors.
Tables should be more straightforward; they have many lines making them confusing.
In figure 2 authors should further explain the process. And in the case of figure 3, the names are just visible to the reader.

General comments

Line 40

Says: Beside

Change to: Besides

Include some frequency values of Chronic liver failure to understand that treating this disease is essential.

Line 68

The abbreviation HAS can be human albumin serum. It seems that the full abbreviation is missing the meaning of the letter S.

Figure 2. It is not clear what happens in each of the circuits. There is a brief description in the text, and there is nothing in the figure for circuits one and two.

Why not put a tube next to an arm of the human schematic? I understand that the first circuit is to pass the blood extracted from the human to the filters. Looking at the drawing, it seems that the sample is taken from his head. For circuit 2, it is not said. It draws that it passes through a 2-4% albumin solution. In figure 2, the third circuit that includes acidification and basification explains that both processes lead to detoxification. However, neither in the figure nor in the figure caption is it said how the flow passes. What must be done first is the correction of metabolic acidosis. The figure shows that two dialyzers are used (acid and basic circuit), but the order of this flow or under what conditions they pass through the two is not specified.

Line 84

I suggest that the authors specify if the sample passes first through the filter that acidifies and then through the one that alkalinizes.

Physiological conformation of albumin is reestablished when blood is first the acidified and then alkalized albumin dialysates pool and the at the end dialysates warm up to body temperature. The purified albumin is then ready for the next cycle.

Physiological conformation of albumin is reestablished when blood is first acidified and then alkalized. Albumin dialysates pool, and at the end, dialysates warm up to body temperature. The purified albumin is then ready for the next cycle.

At the end of the paragraph of line 90, write figure 2

for the next cycle (figure 2).

Line 94

Explain why they suggested citrate, independent of the patient’s prothrombin time.

Table 1

Separate the information and data in the table with fewer lines. The suggestion is to use two sections, one for the four inclusion criteria and the other space for the two exclusion criteria.

Line 122

Quantification of cytokine levels by cytometric bead array

In this section, it is suggested that some references to the methodology used be included. To measure the concentration of cytokines. And it is not only necessary to indicate that the manufacturer's instructions were followed.

Line 123

SAYS: Afterwards, sera were

CHANGE BY: Afterward sera were

Line 159

I suggest clarifying the abbreviation LTX; perhaps it refers to living donor liver transplantation (LTX)

Discussion

There is a paragraph that mentions the limitations of the study. I suggest that the authors include the advantages and disadvantages of the equipment to treat acute, short-term liver failure.

Line 214

SAYS: the size range of 5 to 25 kDA, measuring

CHANGE BY: the size range of 5 to 25 kDa, measuring

Line 233

SAYS: area [26]. Beside immune activation

CHANGE BY: area [26]. Besides immune activation

Table 2 is very confusing.

There are too many lines; at least it should have at least been separated, perhaps with a shadow or a space that would be total patients, cirrhosis etiology, the trigger liver, function how the index is calculated, and the laboratory tests because it is seen as mixed and there are many lines. I suggest that the presentation of table 2 be improved. In particular, the design and the data are acceptable. And not all the data need to be entirely in the center.

Figure 3.

There is no classification; it should also be separated or grouped into which cytokines are anti-inflammatory and pro-inflammatory.

 Inflammatory and cytotoxic because they are all mixed up and do not have a classification or are at least ordered by some pattern to analyze them quickly and understand why they were arranged that way.

Figure 4. The idea of the inflammasome pathway is good since it outlines how the different cytokines are related. What should improve is that the size of the names of the cytokines is reduced IL6 and TNF are not seen; they are well above the circle. Complete TNF by TNFα

It is not commented on because no differences are observed between the cytokine values before and after the process.

From figure 4, the foot of the figure reads.

(Dotted lines indicate higher or lower expression, while solid lines show activation.

In the figure, some dates symbolize increase and decrease. Still, it is difficult to distinguish between the dotted lines, which of them represents a more significant expression from a lesser one.

I suggest using different dotted lines to represent a more significant expression and another separate dotted line showing less expression.

There are several dotted lines to choose from, and this way, it will be easier for the reader to see a line and know what type of expression is showing.

Figure 4

Inside the compartment that represents the core, below NfkB, there are several XXXX, or so they seem. But from the ripple, maybe they're ribosomes. Hint, try to draw something more like a ribosome. And do not superimpose the letters on the circles or figures, as in the case of the word inflammation, which could be seen better if it is placed a little lower. Thus, the lines that cross would not hinder him. If my impression that I have of drawing of the XXXX, they are not ribosomes, that is what I tried to interpret. What does it mean because it is not described in the text or the figure caption? What does it mean? A word can also be included if these crosses represent something.

Reviewer 2 Report

This is my review on “Influence of ADVanced Organ Support (ADVOS) on cytokine 2 levels in patients with acute-on-chronic liver failure (ACLF)” original article.

This study aims to evaluate the cytokine levels before versus after the novel ADVOS treatments on ACLF patients.

Introduction is concise, presenting all the necessary background and establishing the aim of the study.  

Materials and methods are thoroughly described with the appropriate citations. In table 1 the symbol on the 4thbullet on (HRS-AKI) should be included in the footnote section of the table as well (line 109). Exclusion in bold too.

Limitations include my only concern regarding the sample size. However, this is an important report with solid results.

Reviewer 3 Report

Kaps et al. presented an interesting paper concerning evolution of cytokines, representing inflammation, in ADVOS, a new ECAD.

The main limit of this study is the low patients' number, furthermore in a very recent technique.

Also, in this sample, it could be interesting to show invidual evolutions.

Reviewer 4 Report

I read with interest the paper of Kaps et al. 

The paper evaluated the role of the ADVOS on the ACLF and its impact on the cytokine level. 

The paper and the methods are interesting, but I think that to have more reliable information more data are necessary, especially with more than a single treatment. Also, a relation between the cytokine level and clinical improvement would be necessary. 

Round 2

Reviewer 4 Report

Thank you for improving the paper and anwering to the comments.